# OpenReview forum: "Windows Agent Arena: Evaluating Multi-Modal OS Agents at Scale"
_ICLR.cc/2025/Conference — Submitted to ICLR 2025_

### Official Review · Reviewer_wyZz · 2024-10-31

**Soundness:** 2
**Presentation:** 2
**Contribution:** 2
**Rating:** 5
**Confidence:** 4

**Summary:**

This paper introduces WINDOWSAGENTARENA, a benchmark designed to evaluate multi-modal agents interacting with a real Windows OS environment. The primary goal is to address the limitations of existing agent evaluation frameworks (which are often constrained to specific domains or modalities) by providing a scalable and reproducible environment that allows agents to perform a wide range of tasks within the Windows ecosystem. The benchmark includes 150+ tasks across various domains such as document editing, browsing, coding, and system management.

Additionally, the authors propose a new multi-modal agent, Navi, and report its performance on WINDOWSAGENTARENA, achieving a 19.5% success rate. The paper also compares Navi’s performance to human benchmarks and other web-based agent benchmarks like Mind2Web.

**Strengths:**

- The WINDOWSAGENTARENA augments the OSWorld benchmark and extends the evaluation environment to Windows, which is a good complement to the existing benchmarks.
- The authors accelerate the evaluation by enabling parallelized task execution in Azure. This is an important contribution for researchers working with time-consuming multi-step tasks, as it reduces evaluation times from days to minutes.
- The benchmark includes a relatively diverse set of tasks (154 in total), covering different types of user workflows.

**Weaknesses:**

- It looks to me that this benchmark is merely an extension of OSWorld to the Windows environment, with similar tasks, evaluation procedures, and metrics, which limits the contribution of this work. In fact, the benchmark could be further enhanced in the following ways:
    - **Multi-lingual tasks.** The WINDOWSAGENTARENA only covers the tasks and applications in English, but web tasks in different languages are also crucial for comprehensively evaluating the performance of the web agents.
    - **Diverse evaluation metrics.** Currently, the authors only use the success rate as the evaluation metric in OSWorld. Other metrics like the number of steps and abilities like self-reflection could be added.
    - **Cross-application tasks.** All 154 tasks are single-application tasks, yet real-world tasks often include multiple applications.
- The paper compares Navi's performance primarily with human benchmarks. More agents like MM-Navigator or WebAgent are expected to be included.
- The task curation process is not fully transparent. While the authors mention adapting tasks from OSWorld, it is unclear how these tasks were modified for Windows or how the difficulty levels were assigned.

**Questions:**

- [Q1] Please clarify the contribution of this work compared to OSWorld---as the task definition, evaluation procedures, and metrics all follow OSWorld.
- [Q2] Could you provide more detailed qualitative analyses of why Navi fails in certain tasks? For instance, are the failures due to perception errors, action space limitations, or planning issues? This would offer valuable insights for both researchers using the benchmark and those developing future agents.
- [Q3] Why didn’t the paper include a direct comparison with agents from OSWorld (which focuses on Linux)? Even though these benchmarks focus on different OS environments, comparing Navi’s performance on similar tasks (e.g., browsing, coding) could provide more context for the challenges posed by WINDOWSAGENTARENA.
- [Q4] For human performance, an accuracy of 74.5% looks a bit low to me. Could you also provide more detailed qualitative analyses of the failure reasons for human participants? Would it be possible that the given task is infeasible given the current context?

---

> ### Author Response · Authors · 2024-11-14
>
> We thank the reviewer for the valuable feedback. We are encouraged that the reviewer finds our benchmark to complement existing works, and provides a useful parallelization infrastructure. We address each question/concern below:
>
> **Additional benchmark improvements**
> - We agree that benchmark additions like multilingual tasks, improved metrics (e.g. cost, steps, time to completion) can improve agent evaluation. We note that our current tasks and metrics provide ample coverage across the most utilized Windows functions in terms of representative user workflows. Beyond our current analyses of human evaluation (Appendix A.5) like number of steps, difficulty ratings, and agent/task job runtimes (Appendix A.7, Table 11), we are happy to include additional details.
> - We agree; the points raised by the reviewer are improvements that we plan to make over time on the open-sourced codebase nonetheless; this includes improving benchmark tasks in quantity and quality (with a focus on multi-application tasks), adding new applications/domains, new modalities to the benchmark (e.g., audio-capable agents), and adding Chinese, Japanese, and Spanish versions of tasks to our benchmark.
>   - Given that our goal is to first make the benchmark scalable/parallelizable to enable faster evaluation/reiteration, our planned improvements can be easily added atop our framework, along with other features, as we co-develop our benchmark with the research community on Github—especially as model/agentic capabilities evolve.
>
> **Comparison with other agents**
> - Our main goals with Windows Agent Arena are to provide a new benchmark for Windows OS and offer an accessible, preliminary agent (Navi) which researchers can adapt for their own needs. Navi is based on set-of-mark prompting and VLM reasoning, which is a common agent architecture.
> - There is a rich body of literature with multiple agent types, but implementing all these variants was out of scope given our main focus is on the construction of a scalable, parallelizable benchmark for the Windows OS environment. However, we still make a best effort to ablatively study our agent's capabilities as we compare 32 different Navi variants (Table 4) using multiple screen parsing models and VLMs (from small to large models) to help users understand the pros/cons of each configuration.
>   - Our intention for Navi is to illustrate the use of our benchmark and serve as an accessible but relatively capable starting point/template for the community. Together with our benchmark, our hope is that our contribution will enable faster reiteration in terms of research and development of AI agents.
>
> **[Q1] What are our contributions / How is this different from OSWorld?**
> - Please see **[N2]** in "General Response to Common Questions from Reviewers" at the top.
>
> **[Q2] Failure analysis**
> - Section 4.2 details common failure causes for Navi: i) visual-language misalignment (selecting the wrong element ID despite a reasonable textual explanation), and ii) incorrect screen parsing (e.g. grouping multiple elements under the same ID).
> - Figure 6 provides visual examples of these failures. Appendix B (Figure 16) provides another example. We are happy to include more examples in the appendix in the revised manuscript. We hope these examples help researchers when developing their own agents.
>
> **[Q3] Comparison with OSWorld’s agent**
> - OSWorld's agent uses as input the accessibility (a11y) tree along with the raw screenshot. Navi has a similar implementation, but instead uses the Windows UIA tree and other models (e.g., Omniparser) to create SoMs on the screenshot to aid element selection.
> - Moreover, several of our agent tests/choices already share substantial overlap with OSWorld’s including GPT-4V (which delivered the best on-average performance in OSWorld’s experiments and received the most analysis).
> - We also include new models released since OSWorld (e.g., OpenAI’s o1 for planning/reasoning or Omniparser for screen parsing) as well as other types of models not covered by OSWorld or other benchmarks (e.g., small language models like phi3).
> - Overall, we study and analyze 32 variations of Navi (using different screen parsing backbones and VLMs for reasoning/planning).
>
> **[Q4] Human performance**
> - Appendix A.5 provides details on human evaluation/performance.
> - Participants operated without search tools or the internet; failures were mostly due to not knowing the lengthy steps needed to change specific settings in media players and office apps. In several cases, participants were not as familiar with the program/application (e.g., more familiar with another media player). In others, the participants’ typical workflows were more casual which did not require advanced use cases in their day-to-day tasks compared to what the task required. Our tasks intentionally reflect a range of difficulty levels.
>
> Please let us know if there are any other questions and we would be happy to address them!

---

> > ### Comment · Reviewer_wyZz · 2024-11-26
> > **Thank you**
> >
> > Thank you for taking the time to address my comments. However, my concerns regarding W3, Q1, and Q3 still remain. Considering these issues and the feedback from other reviewers, I would like to keep my score.

---

> ### Author Response · Authors · 2024-11-27
>
> We thank the reviewer for the response and we apologize if we missed parts of your inquiry or if our earlier response was unclear. We address your remaining concerns below.
>
> ---
> ## **W3:**
> * Re: task curation details, we refer the reviewer to the section on **“Task Curation for Windows OS“** or **Section 3.2** in our paper.
>   * To summarize: changes include modifying file paths, converting Linux commands into Windows PowerShell, establishing a reverse proxy to communicate with Edge/Chrome via ports on the host machine, revisions to task instructions to match OS/program specifics, etc.
>   * We are happy to expound upon this but this process to modify/adapt tasks is already in the current paper.
> * Re: difficulty ratings, see **Section A.5** of our paper which also contains some detailed analysis.
>   * To summarize: participants rate tasks subjectively. We found that time-based and step-based metrics were susceptible to noise outside of their control (e.g., connection quality, OS updates) and exhibited high variance and disagreement rates, making them uninformative.
>
> ---
> ## **Q1:**
> * Regarding our contributions, we refer to  ***general comment note*** **[N2]** above but highlight/reiterate our key ones below.
> ## Benchmark Content + Tasks
>   * ***Only *10%* of OSWorld's tasks focus on Windows---in contrast, our benchmark focuses exclusively 100% on Windows OS***.
>    * This means that, in absolute terms, **our benchmark has more than *3 times* the number of Windows tasks (>150) than OSWorld (only ~50)**---as a result, we have much more in-depth analysis and more task variety/diversity re: agent performance on Windows.
>      * This is important given Windows is a very widely-used and prevalent OS but less well-represented in agent benchmark environments to see how agents perform with Windows' unique environment. *Even OSWorld's own analysis notes that performance varies significantly between Linux and Windows OS.*
> ## Benchmark Scalability + Accessibility
>   * Our efficient, scalable benchmark infrastructure is cheap and does not require large amounts of storage or GPU resources, etc -- but still *allows for a full evaluation of our benchmark in under an hour with what would normally otherwise take multiple hours or over a day despite the benchmark's size or # tasks*, ***making agent research and development faster and more accessible.*** See general comment **[N2]** above for details.
>   * Agentic benchmarks are typically large/complex due to their need to test various aspects of agentic performance, slowing down evaluation, reiteration, and research. ***Our benchmark does not suffer from that bottleneck.***
>
> ---
> ## **Q3:**
> * We emphasize that our intended key contribution is our benchmark: our primary purpose of providing an agent is to simply demonstrate our benchmark and provide the community with a starting template for experimentation.
> * ***We also point out that comparisons with OSWorld's agent are not practical and, in several cases, not possible.*** Just a few examples:
>   * Some closed-source models used as agents (e.g., OpenAI) by OSWorld have undergone several changes/updates, resulting in changed performance and depreciated models. ***This makes some comparisons impossible (e.g., the version of gpt4-vision-0125 used by OSWorld is depreciated and no longer available).***
>   * Due to differences in the programs/domains we create tasks for in Windows (detailed in our paper) versus those in OSWorld, how success rates are averaged/grouped together by domain will also be different, ***making direct comparisons very difficult and not intuitive.***
>     * Also, running our agent on OSWorld also represents a **significant experiment/request** in light of the amount of time permitted for the discussion period due to OSWorld's sheer size -- **which is also exactly why we believe our benchmark provides a novel contribution in its speedy evaluation and scalability.**
>   * **OS-specific differences:** a11y trees (Linux) and UIA trees (native to Windows) differ in implementation/usage across Windows and Linux due to platform-specific frameworks and conventions. Since these are used as inputs to agents but differ depending on the OS, ***there is no direct comparability***. **In fact, this is why we believe it is important to have a benchmark that focuses on Windows' features.**
>   * Nonetheless, ***a subset of our results already possess some comparability with OSWorld’s agent:*** e.g., the overall performance in our results in **Table 4, Section 4.2** using Omniparser (Set-of-Marks) and UIA trees are comparable/analogous to their counterparts in OSWorld using Set-of-Marks and a11y trees.
>  * We will make revisions in our manuscript to make these points clearer.
>
> ---
> In light of these clarifications, could the reviewer clarify which specific concerns among W3, Q1, and Q3 still remain?
>
> We'd be happy to provide answers, more context, revisions, etc.

---

> > ### Author Response · Authors · 2024-11-28
> >
> > Dear reviewer, we have also just updated our manuscript to reflect several changes. We highlight a few below that specifically relevant to some of your remaining concerns:
> >
> > * Revised introduction and contributions section (pages 1-2) that aims to better frame the problem we're tackling, its context, and clarify our specific contributions
> > * Added section **Appendix A.8 (page 25)** that includes discussion/details on the constraints re: comparability between OSWorld's agent and ours along with additional experiments (Table 12) that attempt to make things more comparable despite fundamental constraints in making apples-to-apples comparison with the OSWorld agent.
> >
> > If you have any other questions or specific concerns among W3, Q1, and Q3 that remain, please let us know. Thank you.

---

> > > ### Author Response · Authors · 2024-12-01
> > >
> > > Dear reviewer, thank you again for your valuable feedback. Since the discussion period is ending soon, we were wondering whether our changes have addressed your concerns. Please let us know and we'd be happy to engage further.

---

### Official Review · Reviewer_jfiZ · 2024-11-03

**Soundness:** 3
**Presentation:** 4
**Contribution:** 2
**Rating:** 5
**Confidence:** 4

**Summary:**

This paper introduced a OS Agent benchmark on Windows OS, and proposed a new MLLM agent called Navi. The goal of this benchmark is to solve the challenges of  restricted modalities and slow process in agent evaluations. The benchmark defined 150+ create 150+ diverse Windows tasks across different domains. Navi is designed with chain-of-thought prompting and tested on this benchmark with different SoMs and MIND2WEB Dataset.

**Strengths:**

The writing of this paper is well. The problem definition is very detailed and easy for the reader to understand. The benchmark is a good complementary to the existing work and the claimed evaluation system is more efficient to the current benchmark system.

**Weaknesses:**

The experimental results lack the comparison of well-known agents, but just use different settings to test on multiple base models. The design of Navi agent is relatively simple and motivation is not clear. The additional experiments on MIND2WEB is not enough to support the agent performance.

**Questions:**

Is Navi just designed as a baseline to this benchmark or an innovative agent method?
Is there a plan to test on more benchmarks for evaluate the performance of Navi in the additional results?

---

> ### Author Response · Authors · 2024-11-14
>
> We thank the reviewer for the valuable feedback. We are encouraged that the reviewer finds that our benchmark is complementary to existing works in the field, and adds a more efficient evaluation. Below, we address each concern in turn:
>
> **Navi’s novelty and Navi’s role in our paper**
> - To clarify, we do not claim in this work that Navi is an entirely new type of agent. Yes, as noted by the reviewer, our goal/aim for Navi (and its variants) was to have it serve as a reasonable, capable, but preliminary starting point and baseline for users, researchers, and the open-source community at large to develop their own agents using our new benchmark.
> - Together with our novel contribution in making this benchmark scalable and parallelizable, we aim to provide an accessible way for the community to train/test agents and reiterate efficiently.
> - Navi follows similar approaches to other web / OS agents which use screen parsing + VLM reasoning. In Table 4 and in the results section of our paper, we provide a thorough analysis of 32 different combinations of Navi agents with distinct backbone models, which can help guide researchers in this area to develop and customize their own solutions.
>
> - We have also open-sourced these backbones to allow researchers to create their own agents which might contain better solutions and move the benchmark forward. In addition, we benchmark Navi on a more established, “traditional” agent benchmark (Mind2web) to verify its performance.
> - Currently, we do not plan on testing Navi on other benchmarks for this paper since our main goal for this paper is not to make Navi competitive. Rather, our paper’s main contribution and novelty are developing a Windows OS benchmark that is scalable through its parallelization infrastructure (for fast evaluation) and realistic for training/testing agents in a Windows environment.
> - To demonstrate the promise of agent development and help showcase the benchmark, Navi serves as both a beginning baseline and openly available template for users to play with and develop in this respect.
>
> We hope our responses have addressed your questions. Please let us know if there are any other questions or suggestions and we would be happy to address them.

---

> > ### Author Response · Authors · 2024-12-01
> >
> > Dear reviewer, thank you again for your valuable feedback. Since the discussion period is ending soon, we were wondering whether our changes have addressed your concerns. Please let us know and we'd be happy to engage further.

---

> > ### Comment · Reviewer_jfiZ · 2024-12-02
> >
> > Thank you for addressing my comments. As you see, Navi is just a baseline, and the main contribution of the work is the benchmark. However, comparisons with SOTA agents are still lacking, and the dataset's innovation is limited. In summary, I will keep my score.

---

> > > ### Author Response · Authors · 2024-12-02
> > > **"SOTA agents**
> > >
> > > We thank the reviewer for taking time to respond to our rebuttal---we very much appreciate it. We further clarify a few points to conclude as well as address remaining concerns or misunderstandings.
> > >
> > > We emphasize that a subset of our extensive experiments and ablations on different versions/variants of Navi *do* indeed include configurations that align with SOTA models/agents (see below for details).
> > >
> > > ## RE: SOTA agent comparisons
> > >
> > > * Many, if not nearly all, well-known agent benchmarks **e.g., [2,3,4,5]** also focus on using the most recent/advanced versions of GPT models at the time of their writing (i.e., GPT-4, GPT-4V, GPT-4o, etc.) as the backbone models, just like we do in our work (GPT-4V, GPT-4o, o1, etc. since our benchmark is multimodal, see Tables 4 and 12 in our paper).
> > > * Some of these other benchmarks/works almost *exclusively* test GPT-series models as agents (e.g., **[4]** from ICLR 2024).
> > > * We also follow standard practice/structuring when testing these backbone models as agents, like these previous works/benchmarks in testing agents, such as action grounding **[1,4,5]**, etc.
> > > * Other benchmarks/works, in addition to the common focus GPT-series models as agents across works, also include a smaller number of other models to test, but they tend to differ from work to work.
> > >    * ***Nonetheless, the best performing agents are almost always those based upon the GPT models (i.e., SOTA)***
> > >    * As such, for our choice of other models, we use Phi 3 and 3.5 given that fewer works look into assessing small multimodal models as backbones for agents.
> > > * In addition to various action/perception grounding methods used by different agents/benchmarks (**Tables 4 & 12** in our paper), we also test Omniparser **[6]** as part of our extensive experiments, which is a very recent work that has shown SOTA performance in improving screen understanding and action grounding (e.g, improving GPT-4V’s performance on Mind2Web, AITW, ScreenSpot, etc).
> > >
> > > Unless the reviewer would like to clarify what else is meant by SOTA agents, we are less clear on this definition and would greatly appreciate the clarification!
> > >
> > > ---
> > >
> > > ## RE: dataset innovation
> > > * We refer to our general comments at the top of this paper for more details.
> > > * To summarize, our innovations and strengths are:
> > >   * Exclusive focus on Windows, providing more than 3x as many tasks as OSWorld across more domains/applications.
> > >   * Extensive tests/ablations on Navi, an agent template we make available as an option to the open-source community.
> > >   * Benchmark scalability, speed/efficiency, and affordability/accessibility (as noted by the reviewer's original comment/review) that is not materially/adversely impacted by increasing the size of the benchmark (to our knowledge, the first work that makes this infrastructure available).
> > >
> > > As always, please let us know of any other remaining questions/comments.
> > >
> > > **References**
> > >
> > > **[1]** "GPT-4V(ision) is a Generalist Web Agent, if Grounded"
> > > * https://arxiv.org/pdf/2401.01614
> > >
> > > **[2]** "ANDROIDWORLD: A Dynamic Benchmarking Environment for Autonomous Agents"
> > > * https://arxiv.org/pdf/2405.14573
> > >
> > > **[3]** "Mind2Web: Towards a Generalist Agent for the Web"
> > > * https://arxiv.org/pdf/2306.06070
> > >
> > > **[4]** "WebArena: A Realistic Web Environment for Building Autonomous Agents"
> > > * https://arxiv.org/pdf/2307.13854
> > >
> > > **[5]** "VisualWebArena: Evaluating Multimodal Agents on Realistic Visually Grounded Web Tasks"
> > > * https://arxiv.org/pdf/2401.13649
> > >
> > > **[6]** "OmniParser for Pure Vision Based GUI Agent"
> > > * https://arxiv.org/pdf/2408.00203

---

### Official Review · Reviewer_qmW7 · 2024-11-05

**Soundness:** 3
**Presentation:** 3
**Contribution:** 2
**Rating:** 6
**Confidence:** 2

**Summary:**

This paper presents WindowsAgentArena, a benchmarking suite designed to evaluate multi-modal agents' abilities within a Windows OS environment. Building on previous work like OSWorld, WindowsAgentArena focuses exclusively on the Windows operating system, making it possible for agents to perform diverse tasks representative of real-world computer usage, such as document editing, web browsing, and multimedia manipulation. The study introduces a novel agent, Navi, which demonstrates the framework’s capabilities, achieving notable results in task performance compared to humans. The benchmark suite also emphasizes scalable and fast evaluation through parallelization on Azure, which reduces evaluation time significantly.

**Strengths:**

1. WindowsAgentArena fills a gap in agent benchmarking by focusing on the Windows OS, which is widely used but less explored in agentic evaluations.
2. The ability to parallelize the evaluation on Azure for faster task completion enhances the framework's practicality and adaptability for large-scale studies.
3. By including tasks across multiple categories (e.g., document editing, web browsing, system management), WindowsAgentArena effectively simulates a real-world computer environment.
4. Detailed performance metrics, examples of successful and failed cases, and analyses of task complexity offer clear insights into the capabilities and limitations of agents.

**Weaknesses:**

1. Windows OS is proprietary, which may limit access for researchers who rely on open-source environments or who cannot accommodate the licensing and resource costs associated with a Windows-based setup.
2. Running Windows virtual machines (VMs) with multiple agents and parallelized tasks can be resource-intensive. This may necessitate high-performance computing setups and large storage capacities, particularly for cloud-based evaluations. Configuring and maintaining the Windows environment, including installing software, handling updates, and ensuring compatibility with the benchmark suite, can also be complex and time-consuming.
3. Agents trained and evaluated solely in a Windows environment may develop strategies tailored to Windows-specific UI elements and workflows, potentially limiting their ability to generalize to other OS environments, computing platforms, or even to updated versions of Windows.
4. WindowsAgentArena builds upon OSWorld’s approach but is specific to the Windows environment. This focus on a different OS may not represent a fundamentally novel leap, as it largely adapts pre-existing methodologies rather than introducing groundbreaking new concepts.
5. Many tasks in WindowsAgentArena replicate basic or common actions found in Windows OS, such as file management or web browsing. The tasks may lack complexity or novelty that could reveal new challenges for agentic behavior, making the approach feel less groundbreaking.
6. The Navi framework leverages established multi-modal approaches, such as UIA tree parsing, DOM tree extraction, and OCR for screen parsing. While effective, these methods are widely used in similar environments, potentially limiting the originality of the agent interaction techniques.
7. The reward structure is based primarily on task completion, similar to other agent benchmarks. WindowsAgentArena does not seem to innovate with intermediate or adaptive reward mechanisms that might encourage more nuanced agent learning behaviors.
8. The framework relies on traditional Set-of-Marks (SoM) and bounding boxes for visual grounding, which are widely used in agent evaluation. Introducing innovative ways to handle visual-language misalignment might have added novelty to the approach.

**Questions:**

See weaknesses.

---

> ### Author Response · Authors · 2024-11-14
>
> We thank the reviewer for the valuable feedback. We are encouraged that the reviewer finds that our benchmark fills a gap in current agent evaluations. Below we attempt to address each of your concerns within the space limit.
>
> **Accessibility and proprietary OS**
> - We are aware of this limitation which is why we have made significant effort so that anyone can access/use Windows OS in our benchmark; despite not being an open-source OS, our code repository enables users on to access a free evaluation copy which can be used for benchmark deployment with easy continued renewal after.
> - To the best of our knowledge, our benchmark is one of the few that provides users open/free access to the Windows OS and computing environment for research/development. Please see **[N1]** in "General Response to Common Questions from Reviewers" Official Comment at the top.
>
> **Resource-intensiveness**
> - See **(2)** of **[N2]** under "General Response to Common Questions from Reviewers" Official Comment.
> - In short, our benchmark does not need any high-performance computing setups and large storage capacities.
> - After each run, the internal state of the system/VM is reset to ensure that progress/completion on one task does not interfere with others. This preserves the integrity of the OS environment so tasks can be continuously run and parallelized without interruption or issue. This also removes all created files etc. resulting from agent action, further mitigating the need for a large storage capacity.
>
> **Configuration/maintenance**
> - Our benchmark and the underlying infrastructure make it easy to install additional software either on the VM directly or via the Docker image. This way, users can design new tasks that require new programs and add them to the benchmark as necessary.
> - Together with the ease in which a user can design their own tasks through a task configuration file/JSON (e.g., see README and Appendix A.6), our benchmark is easily configurable, flexible, and extensible to new tasks, programs, etc., with minimal risk of breaking the benchmark's functionality or integrity.
>   - In defining their own custom tasks, users can leverage our existing library of configuration or evaluation functions or write their own by modeling theirs after ours. We are happy to include even more detailed instructions on task creation, hosting, and configuration beyond what we already have.
> - Unfortunately, many agentic benchmarks with computer/OS environments (e.g., OSWorld, AndroidWorld, etc.) share similar weaknesses when it comes to maintenance: as OS systems undergo refreshes, updates, etc., it can be difficult to ensure continued compatibility with the benchmark.
>   - To mitigate this, we have frozen automatic update and instead conduct manual review/updates (e.g., monthly) to re-adjust and mitigate things breaking. Since we are adding new features, modalities, and tasks, we will be continuously maintaining the benchmark in terms of both our needs and those of the wider community (e.g., Github).
>
> **Ease of benchmark infrastructure**
> - A major contribution of our benchmark is precisely making it easy for researchers to scale up agent experiments. As such, we have made significant effort to automatically build/install all necessary programs, scripts, images, etc. to simplify evals. We also include resources to help add/configure new tasks.
> - Cost-wise, a full parallel run costs in the range of 10 USD in total with CPU machines in Azure. Model calls cost about $100-200 depending on tokens utilized and model choice. We will include these details in our revised manuscript alongside data we already report (Table 10 in Appendix A.7).
>
> **Windows-specificity / Why only Windows?**
> - Please see **[N1]** in the comments we made for all reviewers.
>
> **What are our contributions / How is this different from OSWorld?**
> - Please see **[N2]**.
>
> **Tasks**
> - Please see **[N1]**.
>
> **Navi’s novelty**
> - To clarify, we do not claim that Navi is an entirely new type of agent.
> - We aim for Navi to serve as a reasonable starting point and baseline for researchers developing their own agents.
>   - As pointed out, it follows similar approaches as other web / OS agents which use screen parsing + VLM reasoning.
> - What we provide in Table 4 and in the results section is a thorough analysis of 32 different combinations of Navi agents with distinct backbone models to help guide researchers in this area.
>   - Out of those models we also open-source a few of these backbones to allow researchers to create their own agents which might contain better solutions and move the benchmark forward. We also benchmark Navi on a more well-established “traditional” agent benchmark (Mind2web) to verify its performance.
>
> **Reward structure**
> - We believe that the completion-based reward function is a reasonable assumption for the benchmark. However, we do plan to add number of steps, time, and cost as auxiliary metrics to the benchmark in our open-sourced codebase.

---

### Official Review · Reviewer_ZPBq · 2024-11-06

**Soundness:** 3
**Presentation:** 3
**Contribution:** 2
**Rating:** 5
**Confidence:** 4

**Summary:**

This paper introduces WindowsAgentArena, a benchmark designed to evaluate LLM agents in performing multimodal tasks within the Windows OS environment, including 154 tasks covering office software, web browsing, coding, etc. The benchmark assesses the ability of LLM agents to interact with a complex operating system through various modalities. The paper also provides a baseline agent Navi and test its performance on the benchmark. Codes and deployment guidance are open-source.

**Strengths:**

（1）The paper proposes a new benchmark specifically targeting LLM agents’ performance in complex, real-world tasks on Windows OS. The benchmark is well-designed, with a diverse set of tasks that test multiple aspects of agent capabilities.
（2）The paper provides adequate evaluation on the proposed benchmark with a proposed baseline agent and validates its competitive ability.

**Weaknesses:**

（1）Although the benchmark is claimed to be extensive, it is specifically focused on Windows OS. I’m not sure whether this benchmark could be adapted or extended to other operating systems or environments, as this could broaden its applicability.
（2）Apart from the difference in operating environments, it is unclear how the tasks in this benchmark uniquely assess agent capabilities compared to OSWorld. It seems that the tasks have much overlapping with those in OSWorld. More discussion about key differences is needed to distinguish this work from existing works.
（3）I’m concerned about the usage cost of the benchmark. An API cost estimation for running the tasks in the benchmark is needed.

**Questions:**

（1）What’s the uniqueness of the tasks in the benchmark compared to OSWorld?
（2）Line 263 mentioned that the task procedure is shown in Appendix A.3, but the content in A.3 is about reward evaluation.
（3）Is it possible to extend the benchmark to other operating systems？If the OS difference matters to the agent evaluation, maybe this is necessary

---

> ### Author Response · Authors · 2024-11-14
>
> We thank the reviewer for the valuable feedback and are encouraged that the reviewer finds our benchmark to be well-designed and diverse. We address each concern below in turn:
>
> **Why only Windows?**
> - Please refer to note **[N1]** in the "General Response to Common Questions from Reviewers" Official Comment at the top.
>
> **What are our contributions / How is this different from OSWorld?**
> - Please refer to note **[N2]** in the "General Response to Common Questions from Reviewers".
>
> **Extension to other OS**
> - Generally speaking, there are two aspects/issues to consider if a researcher wants to extend our benchmark to another OS:
>   - Task configurations/evaluation scripts, and benchmark parallelization infrastructure. While OSWorld already, and primarily, covers tasks with Linux OS in mind, it is not yet equipped with fast and scalable parallel evaluation in the cloud. Our approach/environment not only provides benefits when it comes to scalability and faster/parallelizable evaluation but also a more secure environment as well.
>   - Furthermore, OSWorld, to the best of our knowledge, does not natively support a full version of Windows OS to the extent that we do. A developer would be able to borrow infrastructure from Windows Agent Arena, but that project is outside of the scope of our paper. Porting the benchmark to MacOS would require both very heavy engineering work for porting tasks and infrastructure due to the structural differences between OS architectures.
>
> **Uniqueness of tasks and capability assessment**
> - We refer to **[N1]** for more detail. In short, because Windows OS has distinct UI, system-specific functions and command line, system-specific apps, idiosyncrasies, etc. than Linux OS, we expect agentic behavior, performance, and perception to vary considerably---as a result, the same task can have vastly different completion trajectories depending on OS.
>   - As just one example, UIA trees that are used as input (in addition to screenshots and SoMs) to help multi-modal agents understand/access UI elements on the screen exist on Windows; however, different structures exist on Linux, like AP-STI or ATK among others, which naturally will impact agent performance in how well it traverses these structures to perceive different UI/visual elements, etc. These will ultimately impact how the agent plans, reasons, and acts across different OS. We will clarify these aspects in the revised manuscript.
> - Only 10% of OSWorld's tasks are for Windows (43 out of 412) whereas the entirety of our tasks (150+) focus on Windows. Moreover, we include a considerable portion of newly designed tasks in addition to tasks we converted or transformed to suit Windows function and usage. We will add more discussion re: these differences in our manuscript.
> - Rather than focusing on Navi's evaluation across different benchmarks, our main goal is to provide a new benchmark for Windows OS, and offer an accessible, well-performing but preliminary agent (Navi) which researchers can adapt for their own needs. Navi is meant to be a capable starting point and template for the research/open-source community to develop and customize their own agents; together with our benchmark, our hope is that our contribution will enable faster reiteration in the research and development of AI agents.
>
>
>
> **Benchmark cost**
> - Our open-sourced repository README file provides a cost estimate for running the benchmark. This estimate includes both the VM cost (about 8 US dollars for all machines running in parallel for the duration of the evaluation), plus the models' costs (which can vary from 0 US dollars if the user runs a local VLM model, all the way to about 15 US dollars for GPT4o-mini or about 100 US dollars for GPT4o).
> - Appendix A.7 in our paper also includes a table of VM specs/costs and times. For times, we note that typical serial evaluation of a benchmark on the scale of Windows Agent Arena (or larger) would take significantly longer (likely multiple times longer) than our parallel setup.
> - We note that these costs are similar to costs associated with evaluating state-of-the-art closed source models on other agentic benchmarks as well. We also note that as part of benchmark’s infrastructure, we do not need expensive high-performance compute as our parallelization can be done on inexpensive CPU cores (see **[N2]** in the "General Response to Common Questions from Reviewers") so any additional costs from running our benchmark beyond closed-model APIs are near-negligible.
>
> **Mistake in line 263**
> - Thank you for highlighting this mistake. We meant to point the reader to appendix A.6 (TASK DEFINITION & CONFIGURATION). We will change this in our revised manuscript.
>
> We hope our responses have addressed the questions that reviewers have. Please let us know if there are any other questions or suggestions.

---

> > ### Author Response · Authors · 2024-12-01
> >
> > Dear reviewer, thank you again for your valuable feedback. Since the discussion period is ending soon, we were wondering whether our changes have addressed your concerns. Please let us know and we'd be happy to engage further.

---

### Official Review · Reviewer_SbPM · 2024-11-06

**Soundness:** 3
**Presentation:** 3
**Contribution:** 3
**Rating:** 6
**Confidence:** 4

**Summary:**

The paper introduces WindowsAgentArena, a comprehensive benchmarking environment tailored for evaluating multi-modal agents in Windows OS settings. This platform builds upon the OSWorld framework, specifically adapted to provide 154 diverse, complex tasks representative of typical Windows applications and workflows, such as document editing, web browsing, coding, and system customization. The benchmark environment is designed for scalability, allowing for parallelized evaluation on Azure, significantly reducing the time required for agent testing. As a demonstration, the authors develop Navi, a multi-modal agent achieving a 19.5% success rate on WindowsAgentArena tasks, highlighting the challenges in reaching human-level performance (74.5%).

**Strengths:**

1. A new benchmark focuses on multi-modal tasks in the Windows environment.

2. Easy to deploy environment.

3. Authors design an extra multi-modal agent to validate benchmark’s effectiveness

**Weaknesses:**

1. This benchmark does not show clear differences toward OSWorld

2. Agent design can be clearer

3. The number of tasks is limited

**Questions:**

1. For all the designed tasks (Office, Web Browsing, etc), which is specific to the Windows environment? To distinguish WindowsAgentArena from OSWorld, authors need to discuss the unique tasks for Windows.

2. As stated within the paper, two thirds of the tasks are re-implemented from OS World and one third tasks are newly designed. It would be great if authors can show the newly designed tasks to help the audience clearly understand the contribution.

3. What is the backend model for the Agent? Have you tested on different backend models?

---

> ### Author Response · Authors · 2024-11-13
>
> We thank the reviewer for the valuable feedback. We are encouraged that the reviewer finds our work easy to deploy. We address each of your questions/concerns below:
>
> **What are our contributions / How is this different from OSWorld?**
>  - Please refer to **[N2]** in the "General Response to Common Questions from Reviewers" Official Comment at the top.
>
> **Uniqueness of Windows tasks**
>  - Even for web browsing tasks (which are not inherently OS-related), a significant amount of time in terms of engineering and testing was required to modifying the configuration and evaluation functions so that they would work natively on the Windows OS given differences across OS kernels/support.
>     - As just one example for web browsing tasks using Chrome and Edge, functions that access browser cookies, and settings in cached files are usually significantly different between Linux and Windows, and each task had to be modified and verified manually. We also added Edge browser support for several Web tasks, which did not exist in OS World.
> - Additionally approximately double the number of hours was spent creating new tasks for Windows-exclusive apps (Microsoft Paint, File Explorer, Settings, Notepad, Clock). These newly designed tasks are not only meant to work under Windows OS but also aligned with the actions and parts of workflows of common users of Windows---we will better clarify and describe the differences between adapted tasks, converted tasks, and newly designed tasks in Section 3 of our revised paper.
>
> **Number of tasks**
> - With 154 task templates, our benchmark falls in line with major benchmarks in the domain of execution-based evaluation such as MiniWoB++ (114 templates), AndroidWorld (116), Web Arena (241), and OSWorld (369). We are also actively monitoring the open-sourced repository and adding new tasks that we create/source ourselves as well as those made by the research community to expand the benchmark, including users who have already designed their own tasks within our benchmark.
>
> **Agent design back-end**
> - In Table 4 and Section 4, we describe the 32 different agent back-ends we tested along with different combinations of screen parsing and VLM reasoning models. These models range in size and other aspects from small local and cloud models (e.g. Phi3.5 and GPT4o-mini) to large cloud models (GPT4o and o1). We will better clarify and describe the design of our agent Navi in Section 4 of our revised manuscript along with the individual components and their specific considerations.
>
> Please let us know if there are any other questions or suggestions and we'd be happy to answer!

---

> > ### Author Response · Authors · 2024-12-01
> >
> > Dear reviewer, thank you again for your valuable feedback. Since the discussion period is ending soon, we were wondering whether our changes have addressed your concerns. Please let us know and we'd be happy to engage further.

---

### Author Response · Authors · 2024-11-13
**General Response to Common Questions from Reviewers**

We thank the reviewers for their time and greatly appreciate their insightful comments. We are particularly encouraged that all reviewers recognize the impact that a Windows-focused agent benchmark can have. We provide detailed responses to each reviewer’s questions below their individual reviews but also summarize our answers to the most common questions below:

======
- **[N1] Why only Windows?**
  - OSWorld is an agent benchmark primarily focused on agent tasks for non-Windows OS (~90% of OSWorld's tasks are Ubuntu/Linux tasks); Linux has ~4% market share among desktop operational systems. We focus this work exclusively on the Windows OS because, with 73% market share, it is the most widespread OS among PC users.
  - Despite its prevalence, Windows is still relatively under-represented in environments/benchmarks for agentic capabilities. Therefore, OS agents need to include Windows, and make use of its distinct UI, system-specific functions and command line, system-specific apps, and idiosyncrasies.
  - Given its uniqueness, proprietary nature, and under-representation as an OS, it is less clear how agents fare with even common actions/tasks in Windows. Differences between OS software ecosystems can significantly impact an agent’s observation/action spaces and performance, as noted by OSWorld itself (their agent performs ~2x better on Linux than Windows).
    - In our experiments, we see that common actions performed by humans in Windows can be challenging for state-of-the-art models that we use for our agents, suggesting that these tasks, the OS/environment, and the workflow aspects they embody still present specific challenges for agents.

======
- **[N2] What are our contributions / How is this different from OSWorld?**
  - In sum, the key differences and contributions are: **(1) OS focus (Linux vs Windows)** and **(2) parallelization of evaluation (time reduction from days to ~20-30 min, ~100x faster)**.
    - **(1) OS focus**: We mention in our paper that ~2/3 of our tasks are adapted/inspired from OSWorld, and 1/3 are new. There is a non-trivial amount of engineering effort in porting tasks: for each individual task, we rewrite from scratch all configuration functions (which set the initial conditions for the task) and evaluation functions (which verify completeness) so they are compatible not only within a Windows OS but also within our custom parallelizable infrastructure.
      - Section 3.2 understates this effort, which consumed ~800 person-hours for coding, testing, verification, etc. The remaining 1/3 new tasks consumed ~1600 person-hours. This excludes agent testing, etc.
      - **Only ~10% of OSWorld's tasks are specifically for Windows (43/412), We have >3x as many Windows tasks (>150) and focus exclusively on Windows.**
      - Also, despite not being an open-source OS, we made significant efforts to have Windows accessible via our benchmark as a self-contained docker image. We provide users a way to use a free evaluation copy which can be used for benchmark deployment and then easily/continuously renewed thereafter.
      - As a result, our benchmark is one of the few (if not the only) that provides users open/free access to the Windows OS and computing environment for agent research and development. Our benchmark also allows users to install their own programs/applications, add and configure new tasks for their own needs, and more.

    - **(2) Quick/flexible evaluation via benchmark parallelization:** Agent evaluations based on task completion is inherently slow because of the sequential nature of tasks---especially as a benchmark grows in size and complexity.
      - To our knowledge, we are the first to open-source infrastructure which allows researchers to fully parallelize evaluation on our benchmark in under 20-30 min for an arbitrary number of tasks, as opposed to tens of hours or even days or weeks of wait. This enables not only faster evaluation but also faster reiteration in research and development.
      - With our benchmark, there is also *no* need for any high-performance computing setups or large storage capacities. Our setup relies on widely available and relatively low-spec inexpensive CPU cores and does not rely on any GPU resources (Table 10 in Appendix A.7 for details). Our setup also allows users to customize/configure the RAM allocated to the VM---allocated memory can be reduced down to as *small* as **2 GB** and still be able to run the environment.
      - Users can install their own programs onto the image or VM and create their own tasks following details in our paper, README, etc.
      - We also note that our incurred costs are primarily from closed-source model API calls---which are similar to costs incurred from testing closed-source models on other agentic benchmarks as well.

We hope our responses have addressed the common questions that reviewers have. Please let us know if there are any other questions. We thank the reviewers for their time!

---

### Author Response · Authors · 2024-11-28
**Manuscript Revisions**

# Manuscript Revisions

We thank the reviewers for their feedback and have recently uploaded a revised manuscript reflecting reviewers' suggestions, including (but not limited to):

---
## Clarifying Problem-Framing & Contributions
* **Revised introduction section and contribution portion (pages 1-2)** that better frames the problem we're tackling and its context/importance. We also spell out our specific contributions more clearly, such as:
  * ***Windows OS focus & significantly larger number of tasks:*** Our exclusive focus on Windows OS, with a new benchmark containing more than 3 times the number of Windows tasks than OSWorld (>150 vs. ~50), allowing us/researchers to conduct more in-depth analysis of agent performance on Windows---especially since Windows is widely-used but less well-represented in agent benchmark environments.
   * ***Fast + scalable benchmark:***  our benchmark is efficient, scalable, and cheap---and does not require large amounts of storage or GPU resources, etc---but still allows for a full evaluation in less than 0.5 hour with what would normally otherwise take multiple hours or over a day using inexpensive CPU cores, *making agent research/development faster and more accessible.* We are the first to make this sort of benchmark infrastructure readily available.
     * Agentic benchmarks are typically large/complex which can slow down evaluation, reiteration, and research. Our benchmark does not suffer from that bottleneck.
  * ***More/new experiments with agent models:*** extensive experiments and ablations with variants of Navi that include both *popular and less-represented models by similar works* as well as different input processing (Section 4):
    * *Popular models*: large closed-source multi-modal models such as GPT4-V and GPT4-o.
    * *Newer models*: faster models such as GPT4-o mini and slower but more reasoning-capable text-only models like OpenAI's o1
    * *Smaller models*: open-source models like phi-3/phi-3.5 vision.
---
## Additional Results
* **New discussion and results.** We added a new section **Appendix A.8 (page 25)** that includes discussion/details to clarify differences between Windows Agent Arena and OSWorld, including constraints re: comparability between OSWorld's agent and ours as well as results from additional experiments (**Table 12**) that attempt to make things more comparable. Specifically:
  * ***Additional experiments:*** we include additional results with gpt-4v and gpt4-o experiments on our benchmark that use only screenshot or screenshot + UIA tree (Windows accessibility tree analogous to a11y tree) to try and make comparisons possible with some of OSWorld's agent configuration.
    * Combined with our previous results in Table 4, Section 4.2 using Set-of-Marks with UIA trees, these experiments come as close as possible in terms of comparability with OSWorld’s agent due to comparability constraints (see * ***Potential problems in comparisons with OSWorld agents/experiments*** point below).
  * ***Navi***: more discussion/detail on our intended contributions and motivation behind our agent Navi.
  * ***Potential problems in comparisons with OSWorld agents/experiments:*** we discuss issues that unfortunately make direct comparisons misleading and, in some cases, not possible. Examples include but aren't limited to:
    * ***OS-specific differences that affect agent inputs/performance:*** e.g., a11y trees (Linux) and UIA trees (native to Windows) differ in implementation/usage across Windows and Linux due to platform-specific frameworks and conventions. In fact, this is why we believe it is important to have a benchmark that focuses on Windows' features.
    * ***Changes in closed-source models:*** this has resulted in issues with reproducibility and infeasible comparisons (e.g., the version of gpt-4v, gpt4-vision-0125, used by OSWorld is depreciated and no longer available).
    * ***Differences in programs/domains:*** We create many tasks for Windows for programs/domains that do not exist in OSWorld, resulting in both a different number of tasks, domains, and difficulty distributions. As a result, how success rates are scored/averaged/grouped together by domain will differ significantly, as well as overall success rates, and will not be a one-to-one or apples-to-apples comparison.

---

### Meta-Review · Area_Chair_B92v · 2024-12-23

**Metareview:**

This paper presents a Windows-centric extension to OSWorld, with 150 Windows-specific tasks.  On top of that, it provides architecture to run these tasks in parallel, and also on relatively cheap CPU-only hardware + (what I think is Azure-specific?) cloud calls to e.g. OpenAI models.  To validate their benchmark+parallelization approach, they also provide an agent, Navi, that navigates these tasks, with some insights that pop out (e.g., while Navi does pretty well compared to other AI agents, its performance far underperforms humans, and this is likely due to throwing a VLM or VLM+other tricks at a screen to ask for full understanding of state just not quite being there yet).  This is a tricky paper to judge, and that is reflected in the reviewers neutral to neutral-weak responses — on the one hand, this is a comprehensive and useful tool for Windows-oriented developers.  On the other hand, as brought up by nearly every reviewer, this does feel like a clear extension of OSWorld to the Windows ecosystem (+ some tasks), with a straightforward Microsoft-centric parallelized backend for deployment.  That is valuable, but from a novelty point of view reviewers were/are on the fence.

**Additional Comments On Reviewer Discussion:**

Reviewers who responded to the authors' comprehensive rebuttal broadly stayed put on scores.  It is unfortunate that three reviewers did not engage with the rebuttal.  This AC has looked at those reviewers and the rebuttal and is still viewing this as a borderline paper.

---

### Decision · Program_Chairs · 2025-01-22

Reject